

# A new frog of the *Leptodactylus fuscus* species group (Anura: Leptodactylidae), endemic from the South American Gran Chaco

Rosio G. Schneider[1], Dario E. Cardozo[1], Francisco Brusquetti[2], Francisco Kolenc[3], Claudio Borteiro[3], Célio Haddad[4], Nestor G. Basso[5] and Diego Baldo[1]

[1] Laboratorio de Genética Evolutiva, Instituto de Biología Subtropical (CONICET-UNaM), Facultad de Ciencias Exactas Químicas y Naturales, Universidad Nacional de Misiones, Posadas, Misiones, Argentina
[2] Instituto de Investigación Biológica del Paraguay, Asunción, Paraguay
[3] Sección Herpetología, Museo Nacional de Historia Natural, Montevideo, Uruguay
[4] Departamento de Zoologia, Instituto de Biociências e Centro de Aquicultura (CAUNESP). Universidade Estadual Paulista (UNESP), Rio Claro, São Paulo, Brazil
[5] Instituto de Diversidad y Evolución Austral (IDEAUS-CONICET), Puerto Madryn, Chubut, Argentina

Corresponding author
Diego Baldo, diegobaldo@gmail.com

## ABSTRACT

A new species of *Leptodactylus* frog (Anura: Leptodactylidae) from the South American Gran Chaco, morphologically similar and previously confused with the widespread *Leptodactylus mystacinus*, is described through the use of multiple sources of evidence (molecular, external morphology, coloration, osteology, bioacoustics, and behavior). The phylogenetic analysis with partial sequences of mitochondrial rDNA genes (12S and 16S) recovered the new species within the *L. fuscus* group, being highly divergent (>3% genetic distance in 16S). The new species was recovered as sister taxa of *L. mystacinus*, from which it is distinguished by tympanum coloration, cephalic index, dorsum and legs coloration, and some osteological differences in nasals and prevomers. This new frog is characterized by a moderate body size (SVL 46.80–66.21 mm), distinctive color pattern (reddish dorsal surfaces of body with noticeable black stripes in the dorsolateral folds), a circular and dark tympanum with dark tympanic annuli, and behavior of males that call on top of fallen logs and tree branches close to the ground.

## INTRODUCTION

The South American Gran Chaco ecoregion includes portions of northern and central Argentina, southeastern Bolivia, western of Paraguay, and a small area in southwestern Brazil. Temperature and rainfall in this region contribute to an increasing gradient of aridity from East to West, which have led to a distinction between Humid and Dry Chaco subregions (*Dinerstein et al., 1995*; *Olson et al., 2001*). The Dry Chaco occupies an area of nearly 225,500 km², extending over a large part of the northwestern portion of the

ecoregion, being to date the largest most preserved forest/savanna biome in South America, characterized by deciduous forests and arid-semiarid climate with less than 900 mm of annual rainfall (*Grau et al., 2015*).

This ecoregion has a high richness of anurans, with frogs of the genus *Leptodactylus* being one of its most widespread and abundant components (*Köhler, 2000*; *Brusquetti & Lavilla, 2006*; *Álvarez et al., 2009*). This genus comprises 74 species, occurring from southern North America to central South America, including the West Indies (*Frost, 2019*). *Heyer (1969a)*, based on morphology and behavioral characters, assigned *Leptodactylus* species to five phenetic groups: *Leptodactylus fuscus*, *L. marmoratus*, *L. melanonotus*, *L. ocellatus* (=*L. latrans*), and *L. pentadactylus*. The *L. marmoratus* group was later transferred by this same author to *Adenomera* (*Heyer, 1974*), and the remaining four species groups have suffered numerous taxonomic rearrangements (*Heyer, 1969a*, *1969b*, *1970*, *1973*, *1974*, *1978*, *1979*; *Maxson & Heyer, 1988*; *Frost et al., 2006*; *Ponssa, 2008*; *De Sá et al., 2014*).

The monophyly of the *L. fuscus* species group was first tested by *Ponssa (2008)* based on morphological characters. More recently, *De Sá et al. (2014)* provided the first phylogenetic hypothesis for the genus *Leptodactylus* using two different approaches: (a) a molecular matrix (partial sequences of mitochondrial 12S and 16S ribosomal DNA genes, and the nuclear gene Rhodopsin) and (b) these molecular data combined with the morphological character matrix of *Ponssa (2008)*. These authors recovered *Leptodactylus* as monophyletic in the total evidence analyses, but molecular data alone did not support its monophyly (*De Sá et al., 2014*).

The *L. fuscus* species group is the most diverse and widely distributed of the genus, with 30 currently recognized species (*De Sá et al., 2014*) adapted to terrestrial life. Its members have similar external morphology and share some particular reproductive features like oviposition in foam nests that are sheltered in underground chambers, and partial independence of tadpoles from the aquatic environment (*Heyer, 1969b*, *1978*).

*Leptodactylus mystacinus* is a widely distributed species of the *L. fuscus* group that is present in varied environments east of the Andes, in the Yungas of Bolivia and northwestern Argentina, extending eastwards to Paraguay, central and southern Brazil, and more southwards to the Pampas region in central Argentina and Uruguay (*Heyer, Heyer & De Sá, 2003*; *De-Carvalho et al., 2008*). Several works have indicated the existence of some relevant variation among populations of this species along its distribution, concerning morphology and coloration patterns of larvae and adults (*Cei, 1980*; *Heyer, Heyer & De Sá, 2003*; *Langone & De Sá, 2005*), chromosomal polymorphisms (*Bogart, 1974*; *Amaro-Ghilardi et al., 2006*; *Silva et al., 2006*), and genetic divergence for Bolivian lineages (*Jansen et al., 2011*). This variation has led to the proposal of some authors that *L. mystacinus* represents a species complex (*Heyer, Heyer & De Sá, 2003*; *Jansen et al., 2011*; *Jansen, Masurowa & O'Hara, 2016*), which was yet not thoroughly tested.

In the present work we studied specimens of *L. mystacinus* throughout its geographic distribution and found that those specimens from the South American Gran Chaco belong to a different species that is described herein. This new species was distinguished from *L. mystacinus* based on the use of multiple sources of evidence of external adult

morphology, coloration, osteology, behavior, and genetic distances. In addition, we performed a phylogenetic analysis under Maximum Parsimony (MP) and Bayesian approach in order to test its phylogenetic position.

## MATERIALS AND METHODS

### Molecular procedures

We obtained new sequences for 23 specimens previously assigned to *L. mystacinus*, *L. cupreus*, and *L. bufonius*, 20 of which were collected by us, and three obtained from biological collections. Other sequences used in this work come from GenBank (Appendix I).

Total genomic DNA was extracted from ethanol-preserved tissues (muscle/liver) using a saline method (*Aljanabi & Martinez, 1997*). Partial sequences of 12S and 16S ribosomal DNA genes were obtained via the polymerase chain reaction (PCR): a segment of about 800 base pairs (bp) from the 12S rDNA gene was amplified using the primers 12Sa, 12Sb, 12STphef, and 12SRdS (*Kocher et al., 1989*; *Wiens et al., 2005*); while a segment of about 1,050 bp from the 16S rDNA gene was amplified with primers 16sL2A, 16sH10, 16SAr, and 16SBr (*Palumbi et al., 1991*; *Hedges, 1994*). The PCR protocol consisted of an initial denaturation step at 95 °C (10 min), 35 cycles consisting of 95 °C (30 s) for denaturation, 55 °C (1 min) for annealing, and 72 °C (2 min) for extension; and a final extension step at 72 °C (10 min). PCR-amplified products were purified with an Accuprep purification Kit (Bioneer, Oakland, CA, USA) and sequenced in both directions in automatic sequencers ABI 3730XL (Macrogen, Seoul, South Korea). Chromatograms obtained from the automated sequencer were processed using Sequencher v4.5 (Gene Codes, Ann Arbor, MI, USA), and complete sequences were edited with BioEdit v7.0.5.3 (*Hall, 1999*) and deposited in GenBank (Appendix I).

### Phylogenetic analyses and genetic distances

In order to test the distinctiveness and phylogenetic position of the new species, we obtained DNA sequences of two mitochondrial genes (12S–16S » 1,641 bp) from several individuals and localities of this taxon, also of *L. mystacinus* and two additional species of the *L. fuscus* group to complete the sampling: *L. cupreus* and *L. bufonius* (Appendix I). The phylogenetic analyses included sequences of all species in the *L. fuscus* group provided by *De Sá et al. (2014)* available from GenBank (Appendix I). We excluded the 16S sequence of *L. fragilis* employed by these authors (KM091585) because their identity is doubtful (BLAST searches resulted in 92% identity with sequences of *Smilisca baudinii*). We used only the available 12S fragment of *L. fragilis* (KM091469). Additional doubtful or missing sequences were excluded of the analyses, or exchanged for others available in GenBank (see details in Appendix I).

We used as outgroup sequences of all species groups within *Leptodactylus*: *L. bolivianus*, *L. latrans*, *L. macrosternum*, *L. chaquensis* (*L. latrans* group); *L. natalensis*, *L. podicipinus* (*L. melanonotus* group); *L. rugosus*, *L. rhodomystax*, *L. pentadactylus*, *L. myersi* (*L. pentadactylus* group); and we also included *Hydrolaetare caparu*, *Physalaemus cuvieri*, *Engystomops petersi*, and *Hyalinobatrachium fleishmanni* to root the trees (Appendix I).

Sequences were aligned using MAFFT v6.240 (*Katoh & Toh, 2008*) under the strategy G-INS-i and default parameters for gaps opening and extension gaps. The isolated fragments were concatenated using Sequence Matrix v1.8 (*Vaidya, Lohman & Meier, 2011*). Phylogenetic analyses were performed with 1,641 characters, including 68 terminals.

The phylogenetic analyses were performed under MP and Bayesian approach. The first was done with TNT v1.1 (*Goloboff, Farris & Nixon, 2008*), using gaps as missing data, and New Technology search with 1,000 random addition sequences. The branch supports were tested with 1,000 pseudo-replicates obtaining absolute frequencies of Bootstrap and Jackknifing, the latter with removal probability of 36% (*Farris et al., 1996*).

To construct trees under Bayesian approach we firstly used PartitionFinder v1.1.0 (*Lanfear et al., 2012*) to infer the best partition scheme and the evolutionary model that best fitted each partition. We analyzed 12S and 16S genes separately and selected models under the Akaike information criterion (*Akaike, 1973*) using the "greedy" heuristic search algorithm and linked branch lengths. The best model of evolution for both genes was GTR (General time reversible) + I (proportion of invariable sites) + G (gamma distribution). We performed the analysis in MrBayes v3.2.6 (*Ronquist & Huelsenbeck, 2003*) on XSEDE in CIPRES Science Gateway Webserver (*Miller, Pfeiffer & Schwartz, 2010*), implementing the inferred model of nucleotide substitution on two independent runs, each one with four chains sampling every 10,000 generations, for 100 million generations and discarding the first 25,000 trees as burn-in. We verified convergence with TRACER v1.5 (*Rambaut & Drummond, 2007*) and by examining the standard deviation of split frequencies between independent runs ($<0.01$).

For the estimation of sequence divergences within the *L. fuscus* group, including *Leptodactylus* sp. nov. (Appendix I), uncorrected pairwise distances (*p*-distances) of a partial fragment of the mitochondrial 16S rDNA gene were calculated in PAUP* v4.0b10 (*Swofford, 2000*). The sequences were aligned using MAFFT, under the strategy G–INS–i with default parameters for gaps opening and extension, obtaining a matrix of 478 bp.

## External morphology and osteology

Analyzed material, types, and referred specimens, are stored at LGE, IIBP-H, MACN, and CENAI. List of specimens in Appendix II. Institutional abbreviations follow *Sabaj (2016)*. Specimen collections were made in each country with the following authorization numbers: Argentina (MEyRNR 007/2009, 048/2013, 072/2014, 061/2015, 073/2016, 035/2017, DPB 171/2015); Brazil (SISBIO 57098-1); Paraguay (MADES 186/2016, 232/2017, 196/2018); and Uruguay (MGAyP 199/13, 137/16).

A total of 14 morphological measurements were taken from 49 adult specimens of the new taxon (43 males, six females): snout-vent length (SVL), head length (HL), head width (HW), eye diameter (ED), tympanum diameter (TYD), eye-nostril distance (END), interorbital distance (IOD), internarial distance (IND), forearm length (FAL), hand length (HDL), thigh length (THL), tibia length (TL), tarsus length (TSL), and foot length (FL), following *Duellman (1970)* and *Heyer et al. (1990)*. Measurements were recorded with a Mitutoyo digital caliper (0.01 mm). Sex was determined by visual inspection of male

secondary sexual characters and presence of ovarian follicles in females. For comparison, 734 specimens of *L. mystacinus* (584 males, 150 females) were measured.

Osteological descriptions are based on nine adult males (LGE 8085, 15214, 15241, FML 3661, 12315, 12343, 12345–7), previously cleared and stained using the technique of *Taylor & Van Dyke (1985)*. We followed the terminology of *Lynch (1971)*, *Trueb (1973)*, *Emerson (1982)*, and *Ponssa (2008)* for general features; *Trewavas (1933)* for the hyoid and larynx; *Alberch & Gale (1985)* for the phalangeal formula; and *Fabrezi (1992, 1993)* and *Fabrezi & Alberch (1996)* for the carpus and tarsus. For comparison purposes, we cleared and stained three adult males of *L. mystacinus* (LGE 14998, 15207, 15233).

## Advertisement calls

We analyzed 241 advertisement calls of five males (IIBP-H 728, LGE 8085, and three unvouchered specimens) of the new species and 390 advertisement calls obtained from 13 males of *L. mystacinus* (LGE 15209–10, and 11 unvouchered specimens). Advertisement calls were recorded with a Sony WMD6C recorder, and a Sennheiser ME66/K6 directional microphone. The recordings are deposited in the sound collection of the LGE. In addition, we analyzed 850 advertisement calls of 41 males of *L. mystacinus* provided by Fonoteca Neotropical Jacques Vielliard (UNICAMP), Fonoteca Zoológica of the Museo Nacional de Ciencias Naturales (Madrid), and EcoRegistros (Argentina) (details in Appendix III). All recordings were analyzed employing Sound Forge Pro v11.0 (Magix GmbH & Co, Berlin, Germany) with an FFT of 512 points, at a sampling rate of 44.1 kHz and 16-bit precision. The following traits were measured: call duration (miliseconds, ms), intercall interval (ms), call rate (calls/second), and dominant frequency (hertz, Hz), following *Heyer et al. (1990)*.

## Nomenclatural acts

The electronic version of this article in portable document format will represent a published work according to the International Commission on Zoological Nomenclature (ICZN), and hence the new names contained in the electronic version are effectively published under that Code from the electronic edition alone. This published work and the nomenclatural acts it contains have been registered in ZooBank, the online registration system for the ICZN. The ZooBank LSIDs (Life Science Identifiers) can be resolved and the associated information viewed through any standard web browser by appending the LSID to the prefix http://zoobank.org/. The LSID for this publication is: urn:lsid:zoobank.org:pub:7AE8062D-2F27-4F99-AC5E-8D2A94874EF3. The online version of this work is archived and available from the following digital repositories: PeerJ, PubMed Central, and CLOCKSS.

## RESULTS

### Molecular phylogenetic analysis

The MP analysis resulted in four most-parsimonious trees of 3,312 steps (Fig. 1A). The *L. fuscus* group was recovered as sister of a clade formed by the *L. melanonotus* + *L. latrans* groups, but poorly supported. Within the *L. fuscus* group, two large and poorly supported clades were recovered; one that includes *L. ventrimaculatus*, *L. labrosus*, *L. laticeps*, and *L. syphax*, and other that includes the remaining species of the group. The same relationships were recovered in the Bayesian approach (Appendix IV).

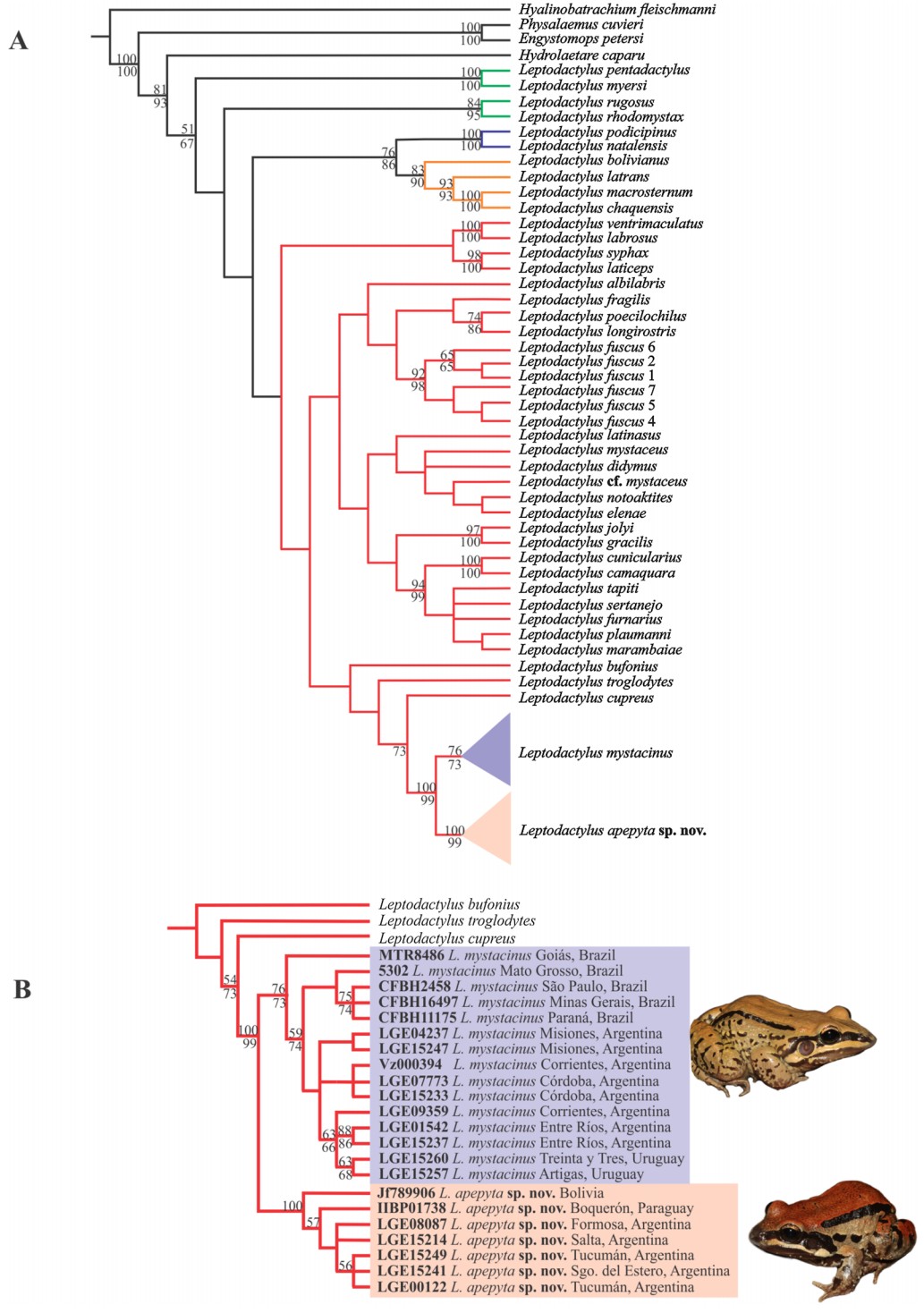

**Figure 1 Consensus tree from Maximum Parsimony (MP) analysis.** (A) Phylogenetic relationships among species in the genus *Leptodactylus*. Branch colors refer to the species groups within *Leptodactylus*: green = *L. pentadactylus* species group, blue = *L. melanonotus* species group, orange = *L. latrans* species group, red = *L. fuscus* species group. (B) Phylogenetic relationships of *Leptodactylus apepyta* sp. nov. and *Leptodactylus mystacinus*, and related species. Bootstrap and Jackknife supports higher to 50% are showing above and below branches, respectively. Photos: Diego Baldo.

Specimens belonging to the *L. mystacinus* species complex were recovered nested in a clade along with *L. cupreus*, *L. troglodytes*, and *L. bufonius*, but forming two independent lineages, strongly supported (Fig. 1B): lineage (I) specimens from the Dry Chaco and surroundings areas, and lineage (II) specimens from the rest of the distribution currently known for *L. mystacinus*. Both lineages have an allopatric geographic distribution and correspond to the two species distinguished in addition by morphology and coloration.

The Bayesian approach showed some subtle differences in topology with respect to the MP analysis (Appendix IV), but in both analyses *Leptodactylus* and the *L. fuscus* group were recovered as monophyletic. However, this relationships of the *L. fuscus* group and the other species groups remain unresolved with this approach. The relationships within the clade containing the new species were consistent with those recovered under MP.

The new species showed significant genetic divergences (*p*-distances) in the 16S rDNA sequences, which ranged 3–4.4% with the closely related *L. mystacinus*, and 5.7–12.5% with the remaining species of the *L. fuscus* group (Appendix V). Furthermore, the new species showed an intra-specific variability up to 1.3%.

## Assignation of available names

***Leptodactylus mystacinus*** (*Burmeister, 1861*)**.** Type locality: "Rozario" (= Rosario, Santa Fe province, Argentina). Type MLU unnumbered, according to *Heyer (1978)*. We examined the type specimen through high quality color photographs (Appendix VI). Although we did not have topotypic specimens to include in our analysis, we examined specimens from nearby localities (Los Nogales and Las Rosas; Santa Fe province; 86 and 95 km, respectively, from Rosario). Their morphology and the location of Rosario, deeply nested within the geographic distribution of lineage II of our phylogenetic analysis, allowed us to assign the name *L. mystacinus* to this widespread lineage.

***Cystignathus labialis*** Cope, 1877**.** Type locality: unknown, "the precise habitat of this species is at present uncertain. It is probably a part of Sumichrast's Mexican collection" (*Cope, 1877*). Type series USNM 31300–5, according to *Kellogg (1932)*. Specimen USNM 31302 considered holotype by *Cochran (1961)*, which is actually a lectotype designation, according to *Frost (2019)*. *Heyer (1978)* studied in detail the type series and considered them conspecific with *L. mystacinus* based on morphology, despite being very faded juvenile specimens. *Heyer (1978)* also discussed the origin of these specimens, noticing that *Cope (1877)* published in that same paper the study of a collection of amphibians and reptiles of origin "unknown, but supposed to be the Argentine Confederation," and proposed that the type series of *C. labialis* could have been part of this collection. High quality photos of four specimens of the type series (USNM 31302, lectotype; and USNM 31303–5, paralectotypes; Appendix VII) of *C. labialis* were analyzed by us. We agree with *Heyer (1978)* that based on their morphology and Cope's original description, they closely resemble *L. mystacinus* sensu lato. Based on the diagnostic characters identified by us (see below), we tried to assign these specimens to one of the two lineages recovered in our phylogenetic analysis. Since the specimens are faded and the diagnostic characters cannot be deduced from Cope's original description, coloration pattern is not useful

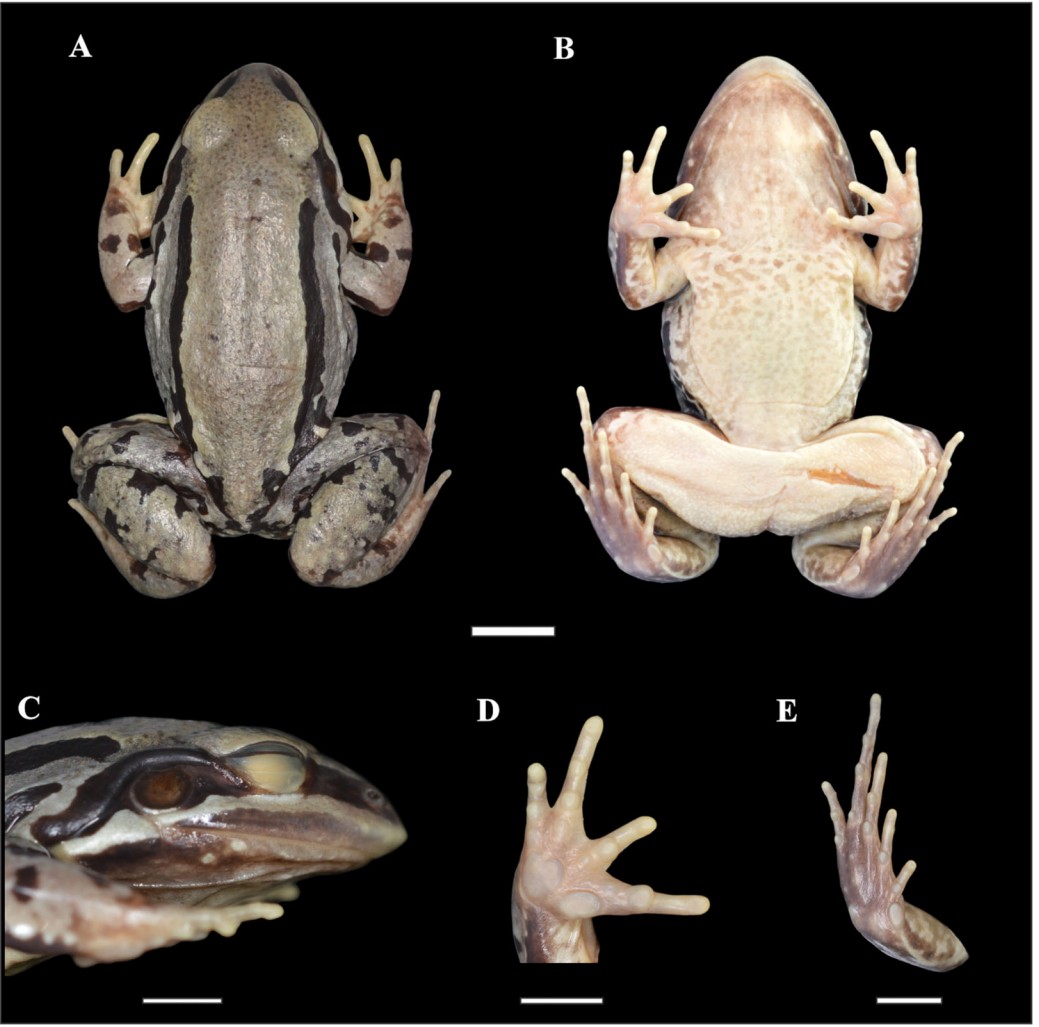

**Figure 2 Holotype of *Leptodactylus apepyta* sp. nov.** (A) Dorsal, (B) ventral, (C) lateral views; (D) ventral view of the right hand, and (E) ventral view of the right foot (LGE 8114). Scale bar = 10 mm. SVL 52.3 mm. Photo: Rosio Schneider.

for this task. Fortunately, their morphometric indexes (Cephalic index = CI = HL/HW, and relation between tympanum and eye diameter = TYD/ED) could be calculated from the photographs, and they overlap with juveniles of *L. mystacinus*, while significantly differ from juveniles of *Leptodactylus* sp. nov.: *C. labialis*: CI 0.98–1.01 (0.99 ± 0.01), TYD/ED 0.51–0.61 (0.57 ± 0.04); *L. mystacinus*: CI 0.95–1.08 (1.01 ± 0.06), TYD/ED 0.49–0.67 (0.59 ± 0.07); *Leptodactylus* sp. nov.: CI 0.80–0.89 (0.84 ± 0.03), 0.43–0.48 (0.45 ± 0.024). Regarding this evidence, *C. labialis* still has to be considered a junior synonym of *L. mystacinus*, leaving lineage I of our phylogenetic analysis without an available name.

## Species description

*Leptodactylus apepyta* sp. nov.

urn:lsid:zoobank.org:act:BCA45181-9C6B-4275-A9E9-BA58D96DE5CE

(Figs. 2 and 3; Table 1)

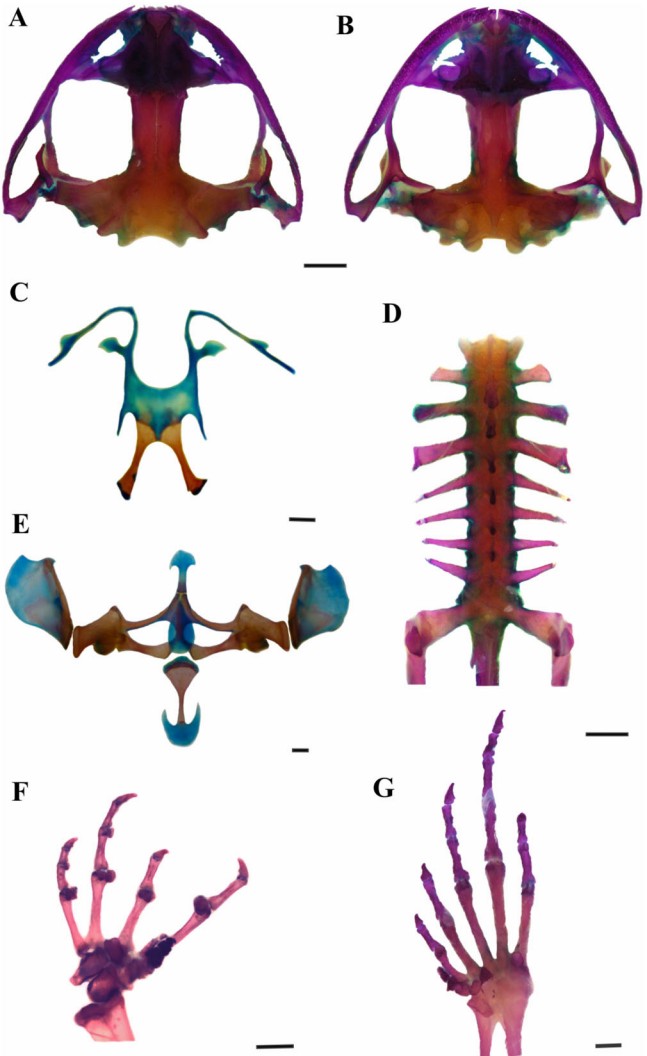

**Figure 3 Skeleton of *Leptodactylus apepyta* sp. nov.** (A) Dorsal and (B) ventral views of skull (LGE 15214), (C) hyoid (FML 3661), (D) vertebral column in dorsal view (LGE 15214), (E) epicoracoid cartilages (FML 3661), (F) palmar view of hand, (G) plantar view of foot (LGE 15214). Scale bar = 2 mm. Photo: Rosio Schneider.

**Chresonymy in Appendix VIII.**

**Holotype.** LGE 8114, adult male, from Argentina, Formosa province, Patiño department, Las Lomitas (24°32′44.1″S, 60°38′33.3″W, 142 m above sea level, asl), collected on January 29, 2014 by Baldo D., Boeris J.M., Brusquetti F., and J. Grosso (Figs. 2 and 3).

**Paratopotypes.** Four adult males: LGE 8085–7, 8113 (same data as holotype).

**Paratypes.** Seventeen adult males: LGE 8181, 8384, 9399, 12290, 12298, 15227, 15236, 15241, 15250–1, 15268, 15272, 15274–6, 15280, 16944; three adult females: LGE 12095, 15240, 15279; and one juvenile: LGE 15232 (details in Appendix II).

**Referred specimens.** Seventeen adult males, five adult females, and 19 juveniles (details in Appendix II).

**Table 1 Morphometric variables (in mm) of adult males and females of *Leptodactylus apepyta* sp. nov. and *L. mystacinus*.**

| Measurements | *Leptodactylus apepyta* sp. nov. | | *Leptodactylus mystacinus* | |
| --- | --- | --- | --- | --- |
| | Males (N = 43) | Females (N = 6) | Males (N = 584) | Females (N = 150) |
| Snout-vent length | 46.80–61.41 (53.77 ± 5.07) | 51.67–66.21 (59.17 ± 5.64) | 45.1–60.87 (51.57 ± 3.18) | 45.47–66.10 (56.25 ± 4.09) |
| Head length | 14.96–21.63 (18.26 ± 1.48) | 18.13–21.55 (20.31 ± 1.55) | 14.96–24.71 (19.44 ± 1.64) | 14.79–27.12 (20.61 ± 2.20) |
| Head width | 15.25–23.46 (20.13 ± 1.89) | 19.97–24.3 (22.06 ± 1.72) | 14.22–22.1 (18.06 ± 1.21) | 16.34–23.18 (19.82 ± 1.50) |
| Eye diameter | 3.67–7.11 (5.60 ± 0.72) | 5.25–6.73 (5.81 ± 0.61) | 2.82–7.37 (5.42 ± 0.63) | 4.04–7.17 (5.70 ± 0.56) |
| Tympanum diameter | 3.08–4.22 (3.74 ± 0.46) | 3.84–4.32 (4.15 ± 0.17) | 3.14–5.87 (4.41 ± 0.53) | 3.65–5.99 (4.76 ± 0.57) |
| Eye-nostril distance | 3.64–5.84 (4.68 ± 0.42) | 4.12–5.50 (4.85 ± 0.56) | 1.95–5.87 (4.69 ± 0.47) | 3.74–6.07 (5.19 ± 0.42) |
| Internarial distance | 2.90–4.64 (3.79 ± 0.46) | 6.65–4.44 (3.99 ± 0.31) | 2.37–4.89 (3.7 ± 0.40) | 2.64–5.27 (4.06 ± 0.46) |
| Interorbital distance | 2.79–4.93 (3.75 ± 0.57) | 3.17–5.32 (4.25 ± 0.91) | 2.27–5.00 (3.56 ± 0.43) | 1.79–5.29 (3.82 ± 0.97) |
| Forearm length | 6.97–12.45 (9.62 ± 1.05) | 9.21–14.31 (11.21 ± 1.73) | 6.41–13.31 (9.59 ± 1.09) | 7.52–14.55 (10.35 ± 1.72) |
| Hand length | 9.80–14.2 (12.44 ± 1.06) | 12.04–13.37 (12.82 ± 0.65) | 8.92–12.3 (12.05 ± 1.26) | 11.00–15.20 (12.97 ± 1.44) |
| Thigh length | 15.76–25.76 (22.42 ± 2.45) | 21.29–26.94 (23.56 ± 2.27) | 11.22–27 (21.59 ± 2.58) | 14.66–28.02 (23.21 ± 5.70) |
| Tibia length | 17.21–25.47 (22.16 ± 1.88) | 22.34–22.57 (24.50 ± 1.86) | 11.87–27.96 (22.38 ± 1.81) | 19.61–30.08 (24.52 ± 3.33) |
| Tarsus length | 9.79–15.43 (12.75 ± 1.32) | 11.75–14.64 (13.50 ± 1.08) | 9.77–24.32 (12.44 ± 1.47) | 10.6–21.72 (13.62 ± 1.76) |
| Foot length | 19.24–25.22 (22.27 ± 1.56) | 22.29–26.03 (24.27 ± 1.45) | 11.08–27.64 (22.86 ± 2.48) | 13.26–28.57 (24.44 ± 2.77) |

**Note:**

N, number of specimens measured. Ranges (Mean ± SD).

## Etymology

The specific epithet is an indeclinable noun, constructed from the words of the Guaraní language *apé* (=back of the neck, dorsum) and *pytã* (=red), in reference to the intense brick red dorsum of adult and juvenile live specimens.

## Definition and diagnosis

*Leptodactylus apepyta* sp. nov. is assigned to the *L. fuscus* group (sensu *Heyer (1969b)*) by its phylogenetic position, and by the presence of the following synaphomorphies (*Ponssa, 2008*): (1) tectum nasi and alary process of premaxilla at the same level; (2) frontoparietal with posterior margin convex, and (3) cultriform process of parasphenoid sited between neopalatines.

The new species is diagnosed within the *L. fuscus* group by the following combination of character states: (1) moderate size sensu *Heyer & Thompson (2000)* (SVL 46.80–61.41 mm in males; 51.67–66.21 mm in females); (2) robust body aspect in dorsal view; (3) head wider than long (CI 0.77–0.95); (4) small, circular, and dark tympanum, with dark tympanic annuli; (5) black broad stripe from tip of snout to the insertion of the forelimb; (6) a distinct light upper lip stripe; (7) one or two pairs of dorsolateral folds, with distinct uninterrupted dark stripes coincident with the upper pair, and interrupted dark stripes in the dorsolateral folds of the flanks; (8) reddish color on dorsal surfaces of body and limbs; (9) dorsum with small dark spots; (10) thigh, tibia, and tarsus with broad, diffuse, and dark bars; and (11) advertisement call composed by a single, short (30–68 ms), and non-pulsed note; call rate of 3.86–7.69 calls/s, without harmonic structure and with dominant frequency between 2,155 and 2,457 Hz; (12) males usually call on the top of fallen logs and low branches of trees.

## Comparisons with other species

According to the available information, *L. apepyta* sp. nov. can be distinguished from all other known species of the *L. fuscus* group by a combination of external morphology, osteology, acoustics, behavior, and rDNA sequences (12S and 16S).

*Leptodactylus apepyta* sp. nov. is distinguished from *L. albilabris*, *L. latinasus*, *L. longirostris*, *L. marambaiae*, *L. caatingae*, *L. camaquara*, *L. cunicularius*, *L. furnarius*, *L. tapiti*, *L. elenae*, *L. fragilis*, *L. oreomantis*, and *L. plaumanni* by the larger size (SVL 46.80–61.41 mm in males; 51.67–66.21 in females of *L. apepyta* sp. nov.; combined SVL 23–46.4 mm in males; 29.1–49.6 mm in females of the other species; *Heyer, 1978*; *Sazima & Bokermann, 1978*; *Heyer & Heyer, 2002*; *Heyer & Juncá, 2003*; *Heyer, Heyer & De Sá, 2006*; *De Carvalho, Leite & Pezzuti, 2013*; *De Sá et al., 2014*) and from *L. laticeps* by the smaller size (94.2–109.7 mm in males; 88–117 mm in females; *De Sá et al., 2014*).

*Leptodactylus apepyta* sp. nov. can additionally be distinguished from *L. camaquara*, *L. furnarius*, *L. jolyi*, *L. tapiti*, *L. fuscus*, *L. gracilis*, *L. longirostris*, *L. marambaiae*, *L. oreomantis*, *L. plaumanni*, *L. poecilochilus*, and *L. sertanejo* by the absence of a longitudinal mid-dorsal light stripe, present in the other mentioned species (*Heyer, 1970*, *1978*; *Sazima & Bokermann, 1978*; *Giaretta & Costa, 2007*; *De Carvalho, Leite & Pezzuti, 2013*, *De Sá et al., 2014*). The presence of one or two pairs of dorsolateral folds in the new species differentiates it from *L. bufonius*, *L. latinasus*, *L. troglodytes*, *L. fragilis*, *L. laticeps*, *L. syphax*, and *L. cunicularis*, with dorsolateral folds undefined or absent (*Heyer, 1978*; *Heyer, Heyer & De Sá, 2006*, *2010*; *Sazima & Bokermann, 1978*; *De Sá et al., 2014*). A light colored line on the dorsal surface of the tibia separate *L. albilabris*, *L. caatingae*, *L. cupreus* and members of the *L. mystaceus* complex –*L. elenae*, *L. mystaceus*, *L. notoaktites*, *L. spixi*, and *L. didymus*– (*Heyer, 1978*, *1983*; *Heyer & Juncá, 2003*; *Heyer, García-Lopez & Cardoso, 1996*; *Caramaschi, Feio & São-Pedro, 2008*) from the new species, where this line is absent. The distinct light upper lip stripe differentiates the new species from *L. labrosus* and *L. ventrimaculatus*, which lack distinct stripe (*Heyer, 1978*).

*Leptodactylus apepyta* sp. nov. differs from the closely related *L. mystacinus* (character states in parentheses) by: tympanum with dark annuli (light-colored annuli); head wider than long, CI 0.77–0.95 (head longer than wide, CI 0.99–1.26); the dorsum generally with pronounced reddish coloration with small dark spots (brown coloration); and legs with wide and diffuse stripes (thin and defined stripes) (*Heyer, 1978*; *Heyer, Heyer & De Sá, 2003*). There are also some subtle osteological differences between *L. apepyta* sp. nov. and *L. mystacinus* (Fig. 3 and Appendix IX): the nasals in *L. mystacinus* are triangular with proportions equivalent in length and width, while in *L. apepyta* sp. nov. they are wider, stretched, with slightly concave outer edges. The lateral edges of the middle ramus of prevomers of *L. apepyta* sp. nov. are irregular, more than in *L. mystacinus* in which they appear smooth giving it a triangular aspect.

The advertisement calls of *L. apepyta* sp. nov. (Figs. 4A and 4B; Table 2) consist of a single type of non-pulsed note and separates it from *L. albilabris*, *L. caatingae*, *L. gracilis*, *L. jolyi*, *L. mystaceus*, and *L. sertanejo* which produce pulsed calls (*Heyer, 1978*; *Sazima & Bokermann, 1978*; *Heyer & Juncá, 2003*; *Giaretta & Costa, 2007*), and from *L. cunicularius*,

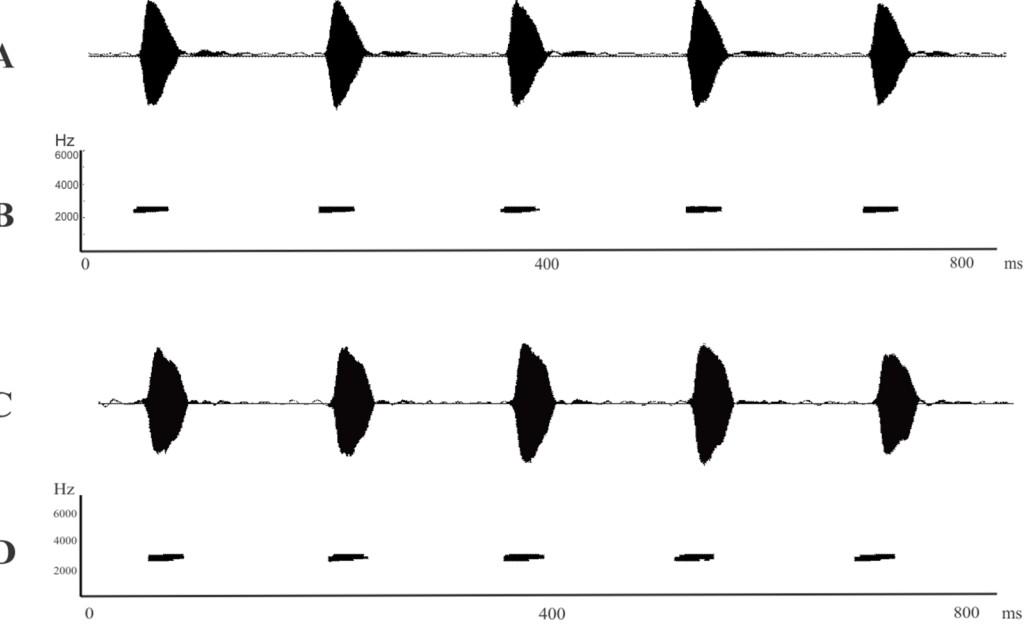

**Figure 4 Advertisement calls.** (A) Oscillogram and (B) spectrogram of *Leptodactylus apepyta* sp. nov. from Las Lomitas, Formosa province, Argentina (LGE 8085); (C) Oscillogram and (D) spectrogram of *Leptodactylus mystacinus* from Rosario, Santa Fe province, Argentina (unvouchered specimen). Miliseconds (ms); Hertz (Hz).

**Table 2 Advertisement call variables of *Leptodactylus apepyta* sp. nov. and *L. mystacinus*.**

| Bioacoustics variables | *Leptodactylus apepyta* sp. nov. N = 5 (241) | *Leptodactylus mystacinus* N = 41 (1,250) |
|---|---|---|
| Call duration (ms) | 30–68 (42.2 ± 0.82) | 27–66 (43.62 ± 0.74) |
| Intercall interval (ms) | 88–204 (132.5 ± 2.91) | 84–221 (124.60 ± 2.31) |
| Dominant frequency (Hz) | 2,155–2,457 (2,266.04 ± 80.41) | 1,960–2,874 (2,368.88 ± 195.05) |
| Calls/second | 3.86–7.69 (5.92 ± 1.04) | 3.94–8.40 (6.08 ± 0.91) |

**Note:**
    *N*, number of recorded individuals (number of analyzed calls). Ranges (Mean ± SD).

*L. oreomantis*, and *L. plaumanni*, with trilled calls (*Sazima & Bokermann, 1978*; *Kwet, Di-Bernardo & Garcia, 2001*; *De Carvalho, Leite & Pezzuti, 2013*). In comparison with the remaining species in the *L. fuscus* group that also have calls with non-pulsed notes, the dominant frequency of the new taxon (2,155–2,457 Hz) is higher than those of *L. bufonius*, *L. dydimus*, *L elenae*, *L. fragilis*, *L. notoaktites*, *L. poecilochilus*, and *L. spixi* (combined dominant frequency range from 470 to 2,033 Hz; *Heyer, 1978*; *Heyer, García-Lopez & Cardoso, 1996*; *Bilate et al., 2006*); and lower than those of *L. cupreus*, *L. furnarius*, *L. latinasus*, *L. marambaiae*, and *L. tapiti* (combined dominant frequency range from 2,600 to 4,000 Hz; *Heyer, 1978*; *Sazima & Bokermann, 1978*; *Caramaschi, Feio & São-Pedro, 2008*; *Brandão et al., 2013*).

The call duration of *L. apepyta* sp. nov. ranges from 30 to 68 ms, being shorter than those of *L. fuscus* and *L. camaquara* (150–300 ms; *Sazima & Bokermann, 1978*; *Heyer &*

*Reid, 2003*). It can also be distinguished from *L. longirostris* by a faster call rate (5.92 *vs.* 1.4–2.0 calls/s, respectively; *Crombie & Heyer, 1983*).

*Leptodactylus apepyta* sp. nov. and *L. mystacinus* cannot be distinguished from each other by their advertisement call, both species have similar structure and overlapping measured variables (Fig. 4; Table 2). The advertisement calls of *L. mystacinus* (Figs. 4C and 4D), consist of a series of single non-pulsed notes, produced at a rate of 6.08 calls/s. Call duration ranges from 27 to 66 ms (43.62 ± 0.74 ms), and intercall interval ranges from 84 to 221 ms (124.60 ± 2.31 ms). The advertisement call lacks amplitude and frequency modulation, and dominant frequency ranges from 1,960 to 2,874 Hz (2,368.88 ± 195.05 Hz).

## Description of the holotype

Adult male. Robust body aspect, moderate size (SVL 52.30 mm), head wider than long (CI 0.9), HL 34% of SVL. Snout sub-elliptical in dorsal view, and protruding in lateral view. Canthus rostralis indistinct, loreal region oblique. Nostril closer to tip of snout than to the eye. Prominent eye located laterally. Upper lip with a distinct light stripe; black broad stripe from tip of snout to the insertion of the forelimb, passing on the eye and tympanum. Tongue large and free posteriorly, notched from behind. Vomerine and maxillary teeth present. Tympanum evident, circular with dark annuli, largely separated from eye. Tympanum diameter smaller than eye diameter (TYD/ED 0.7). Supratympanic fold developed, from the eye to the forelimb insertion. Commissural gland present. Dorsum with small dark spots. Two pairs of dorsolateral folds. The upper pair accompanied by wide uninterrupted dark stripes; and interrupted dark stripes in the folds of the flanks. Flanks with small-medium dark spots. Skin rough, with small white tubercles on dorsum and flanks. Belly spotted. Dark throats, vocal sacs evident. Cloacal zone without tubercles, femoral zone with smooth texture. Arm robust, stained dorsally. Hand with slender fingers, with rounded tips, without webs or fringes. Relative finger lengths IV > II = V > III. Subarticular tubercles rounded, outer metacarpal tubercle rounded, inner metacarpal tubercle elongate. No thumb asperities or prepollex visible. Legs robust, tibia larger than thigh; broad and diffuse bars in thigh, tibia, and tarsus. Toe tips rounded, without webs or fringes. Relative toe lengths IV > III > V > II > I. Subarticular tubercles rounded, outer metatarsal tubercle small and rounded, inner metatarsal tubercle large. Small white tubercles in the sole of tarsus.

Measurements of the holotype (in mm): SVL 52.30; HL 17.60; HW 19.70; ED 5.70; TYD 4.00; END 4.50; IOD 3.60; IND 3.90; FAL 8.70; HDL 12.40; THL 20.90; TL 21.40; TSL 12.00; FL 21.10.

## Coloration in life

Dorsum uniformly reddish (Fig. 5A). A wide black stripe from the tip of snout to the supratympanic fold. Black tympanic membrane, with black tympanic annulus. Near the upper half of the eye brown, and black below. Upper and lower lips dark gray. Dorsum with small black spots. Black stripes accompany the dorsolateral folds, spreading in the dorsum, from the nearest of supratympanic fold to groin. Forelimb reddish above, with a black stripe on anterior and posterior sides of arm and diffuse black stripes on forearm.

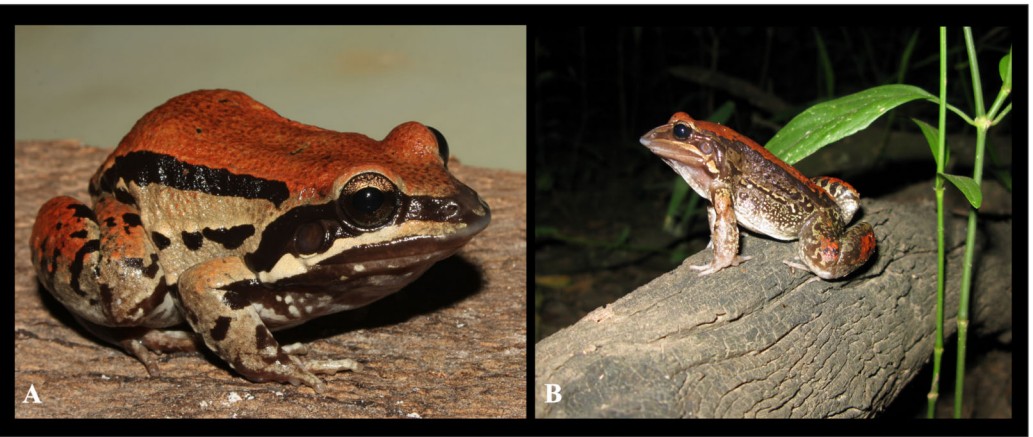

**Figure 5 *Leptodactylus apepyta* sp. nov.** (A) Dorsolateral view of the holotype (LGE 8114) in life; (B) male vocalizing from a log (LGE 15240). Photo: Diego Baldo.

Thigh reddish above, with wide, incomplete and diffuse dark bars in thigh and tibia. Belly and ventral surfaces of forelimb and leg clear cream with diffuse spots. Gular region dark. Cloacal zone with distinct white tubercles.

## Color in preservative

The reddish coloration becomes gray. Black stripes that accompany dorsolateral fold, lateral head stripe, arm, and leg maintained the dark coloration. The belly continued whitish and the vocal region grayish. Tympanum kept its dark coloration, with dark tympanic membrane and dark tympanic annul. Eyes become complete black.

## Variation

The reddish coloration of the dorsum of *L. apepyta* sp. nov. can be brightly red (LGE 8087, 8181, 15274–5, 15280, 16944, IIBP-H 728–9, 1738), sometimes less marked (LGE 122, 8085–6, 8113, 8384, 9399, 11231–3, 12095, 12290, 12298, 15214, 15227, 15232, 15236, 15238, 15240–1, 15248–51, 15268, 15272, 15276, 15279, IIBP-H 2308, 2848–9) (Fig. 6). In *L. mystacinus*, a vast variation in coloration patterns was observed; most specimens have different shades of brown coloration (Fig. 7). The specimen LGE 7890 of *L. mystacinus* from Argentina, Misiones province, Itacaruaré showed a reddish coloration, similar to that of *L. apepyta* sp. nov., but it was clearly identifiable as *L. mystacinus* according to other morphological characteristics, e.g. tympanum coloration (Fig. 7H). Some specimens have thinner and conspicuous stripes in the legs (LGE 8113, 16944), while others have diffuse or absent stripes (LGE 8086, 8384, 12290, 15263). The folds on the flanks are incomplete or absent. Specimen LGE 15251 has an oblique stripe that goes through the dorsum. The specimens LGE 8384 and 15250 have the posterior portion of the tympanic annuli less pigmented. Females are larger than males (SVL females 59.17 ± 5.64 in $N = 6$; males 53.77 ± 5.07, $N = 43$), and the males have brownish or blackish vocal sacs. Specimen LGE 15232 has a malformation consisting of a duplication of the last phalange of the fourth toe.

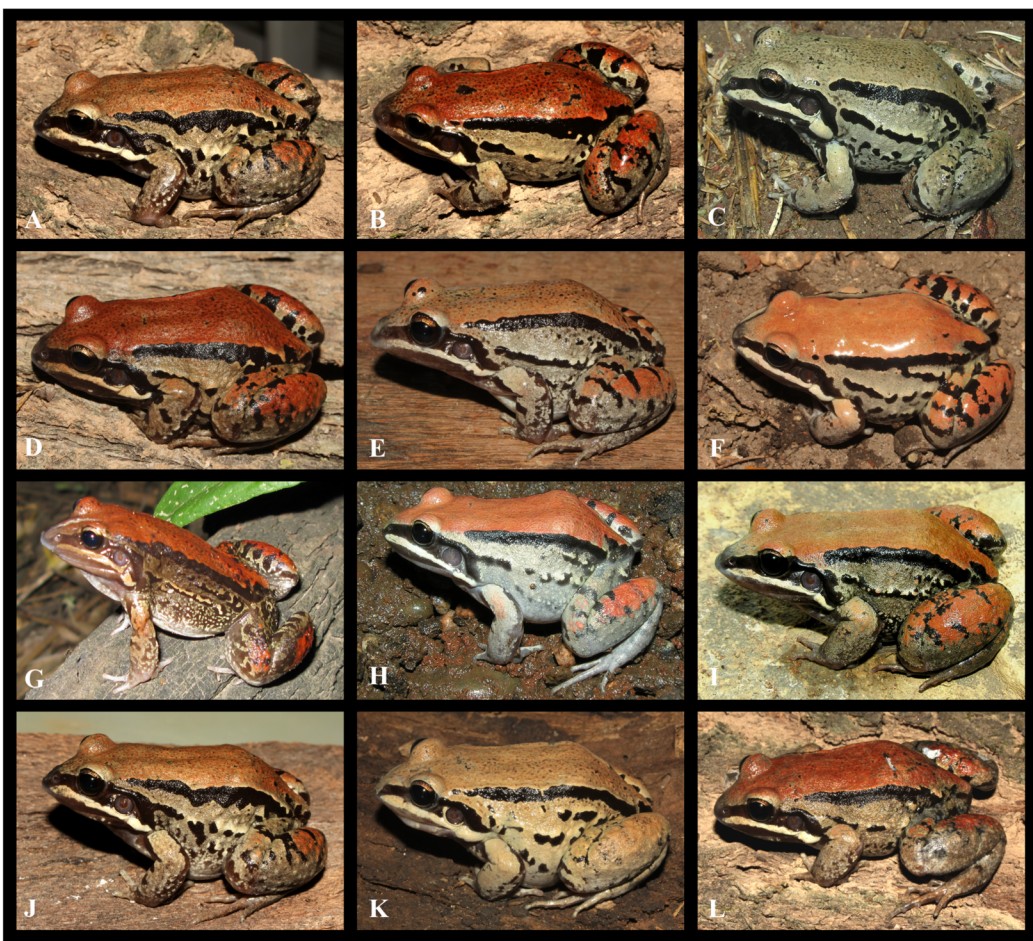

**Figure 6 Intraspecific variation observed in *Leptodactylus apepyta* sp. nov.** (A) LGE 8085, (B) LGE 8087, (C) LGE15241, (D) LGE 8114, (E) LGE 8181, (F) LGE 9399, (G) LGE 15240, (H) LGE 15232, (I) IIBP-H 729, (J) LGE 8113, (K) LGE 8384, (L) LGE 8086. See Appendix II for locality data. Photos: Diego Baldo.

## Osteology

Skull wider than long (Figs. 3A and 3B). Sphenethmoid with lateral margins straight. Prootics and exoccipitals fused, with protuberant crest noticeable, condyles widely separated. Frontoparietals rectangular that do not reach nasals. Anterior borders irregular, lateral borders parallel, posterior margin convex, overlapping the prootics. Nasals expanded and subtriangular, in contact along their entire length, not overlapping the maxillary. Laterals border irregular and curved. Nasal capsule separated by the nasal cartilage. Parasphenoid t-shaped, cultriform process serrated anteriorly, between neopalatines. Neopalatines posteriorly concave, overlap sphenethmoid in their inner portion, and contact maxilla in the outer edge. Prevomers in broad median contact and articulate the maxilla anteriorly. Middle ramus of prevomers bifid and rectangular, dentigerous process curved with 13–17 teeth each. Complete maxillary arch. Premaxillas not fused, with 8–10 conical, curved and bicuspid teeth each. Alary process subrectangular. Pars palatina subrectangular, palatine process bifid. Maxillas bearing 47–51 teeth

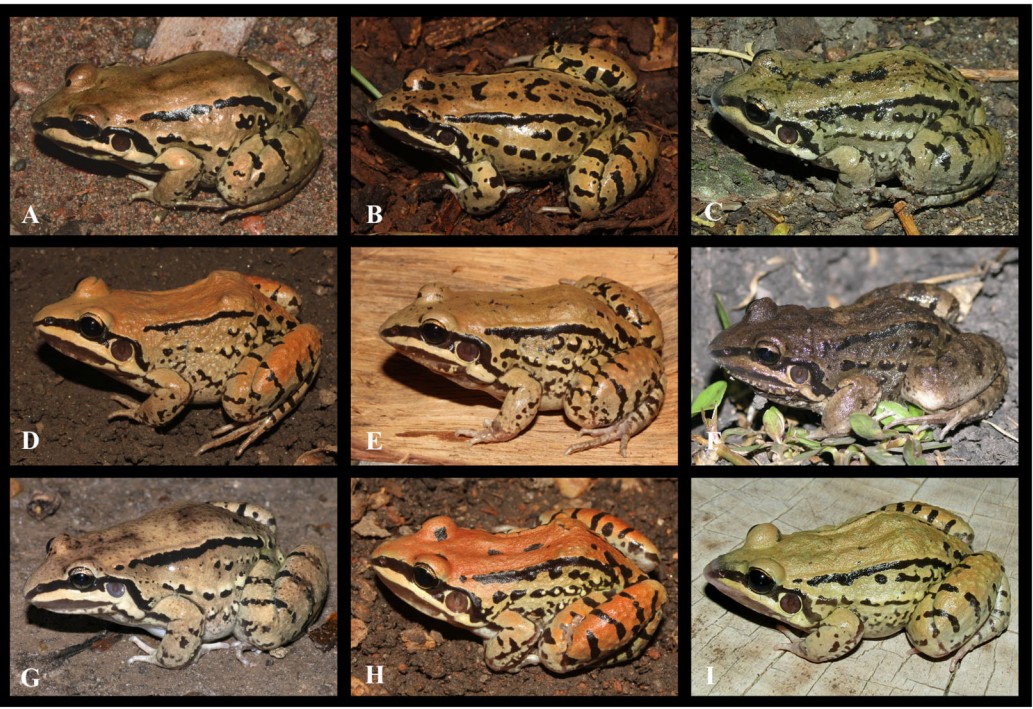

**Figure 7 Intraspecific variation observed in *Leptodactylus mystacinus*.** (A) LGE 20956, (B) LGE 7773, (C) LGE 15206, (D) LGE 7606, (E) LGE 9343, (F) LGE 15254, (G) LGE 15264, (H) LGE 7890, (I) IIBP-H 1086. See Appendix II for locality data. Photos: Diego Baldo.

each, anterior tip sharp and overlaps the premaxilla, posterior end articulates with quadratojugal. Quadratojugals well developed, contacting maxilla. Mentomeckelian L-shaped. Dentary curved anteriorly. Angular with anterior end sharp and articular region mineralized. Pterygoids in contact with parasphenoid and articulate with the maxilla, anterior ramus expanded anteriorly; medial ramus reaches the prootic and posterior ramus laminar and curved. Squamosals with zygomatic ramus subtriangular and curved, otic ramus subtriangular and descendent ramus with canal in their middle extension. Hyoid plate wide and cartilaginous (Fig. 3C). Hyale processes thin and curved. Alary process thin, perpendicular to the hyoid plate. Posterolateral processes thin and mineralized. Posteromedial processes mineralized, with posterior ends cartilaginous. Larynx with triangular arytenoids and thinner cricoids as a circular ring. Vertebral column composed of eight procoelous, imbricate, presacral vertebrae (Fig. 3D). Atlas not fused to the first vertebra, with centrum wider than other vertebrae in ventral view. Cervical cotyles widely separated (type I of *Lynch, 1971*); semilunar lateral and anteriorly oriented. Intercotylar region concave. Relative lengths of transverse processes III > IV > V = VI = VII = VIII > II. Parasagittal processes slightly curved, neural arch thin and sharp. Relative lengths of vertebral centra 2 < 3 < 4 = rest of vertebrae. Sacrum with modified transverse processes, sacral diapophyses oriented toward the back, wider than presacral vertebrae. Ilio-sacral articulation short, narrow internal ligament joins on the distal part of sacral diapophysis (type IIB of *Emerson, 1982*). Sesamoid of the sacral diapophyses present. Sacral-coccygeal

articulation bicondylar. Urostyle cylindrical, wide in the anterior portion, thinner towards its posterior end, with notorious dorsal crest.

Pectoral girdle arciferal (Fig. 3E). Episternum cartilaginous and stick shaped. Omosternum cartilaginous, distal end expanded. Xiphisternum expanded, mineralized anteriorly, the proportions between width and length vary according to the individual. Mesosternum ossified, with medium line and anterior edge cartilaginous. Epicoracoid cartilages broadly overlapped (right over the left). Clavicles curved and separated. Scapulas rectangular shaped and wider than clavicles, with pars acromialis larger than pars glenoidalis. Coracoids subrectangular, with distally expanded end, wider than clavicles. Cleithrum mineralized. Supraescapulas cartilaginous and mineralized, rectangular shaped and wide in the distal part, with variation in size according to the individual. Humerus slightly curved. Caput humeri rounded. Eminentia capitata expanded and rather flattened. Humeral crista notorious. Radius-ulna fused. Carpal morphology with ulnare, distal carpal 5–4–3, element Y, distal carpal 2, radial and proximal element of prepollex (type E of *Fabrezi, 1992*; Fig. 3F). Phalangeal formula 2–2–3–3 and relative length of digits IV > II = V > III. Fingertips knobbed. Glide and palmar sesamoids. Prepollex with three segments. Nuptial spines absent. Pelvic girdle. Ilium with preacetabular area projects anteriorly as a wedge, rounded border with well-defined curved edges and ilial shaft stick shaped with well-developed dorsal crest. Ischium with postacetabular expansion triangular. Pubis mineralized as a wedge between ilium and ischium. Femur slightly curved. Tibio-fibula equal to femur in length, cartilage sesamoid present. Tarsus composed of element Y, distal tarsal 1 and distal tarsal 2–3 (Fig. 3G). Glide and plantar sesamoids present. Prehallux composed of three segments. Phalangeal formula 2–2–3–4–3, toe tips knobbed.

## Advertisement call and natural history notes

*Leptodactylus apepyta* sp. nov. is a species typical of forests and open areas of the Dry Chaco's environments, characterized by scarce rainfalls and deciduous vegetation. Males were found vocalizing after sunset during the rainy season, from mid-October to February. They usually begin to vocalize near to temporary ponds on the top of fallen logs and low branches of trees, up to 1.5 m high (Fig. 5B). The advertisement call consists of series of non-pulsed notes emitted at a rate of 5.92 notes/s (Figs. 4A and 4B; Table 2). Note duration ranges from 30 to 68 ms (42.2 ± 0.82 ms), and internote interval from 88 to 204 ms (132.5 ± 2.91 ms). The call lacks amplitude and frequency modulation. Dominant frequency = fundamental frequency, ranges from 2,155 to 2,457 Hz (mean 2,266.04 ± 80.41 Hz). Oviposition and tadpoles are unknown.

## Geographic distribution

*Leptodactylus apepyta* sp. nov. occurs in the South American Gran Chaco in Argentina and Paraguay (Fig. 8), almost exclusively inhabiting the Dry Chaco, with some scarce records in surrounding areas of Humid Chaco (i.e., Pirané, Formosa province, Argentina; and Concepción, Paraguay) and the Yungas ecoregion (i.e., Zapla, Jujuy province, Argentina). Additionally, a GenBank sequence (JF789906, voucher not examined by us) of a specimen from the Chiquitanía ecoregion (Ñuflo de Chávez province, Santa Cruz department,

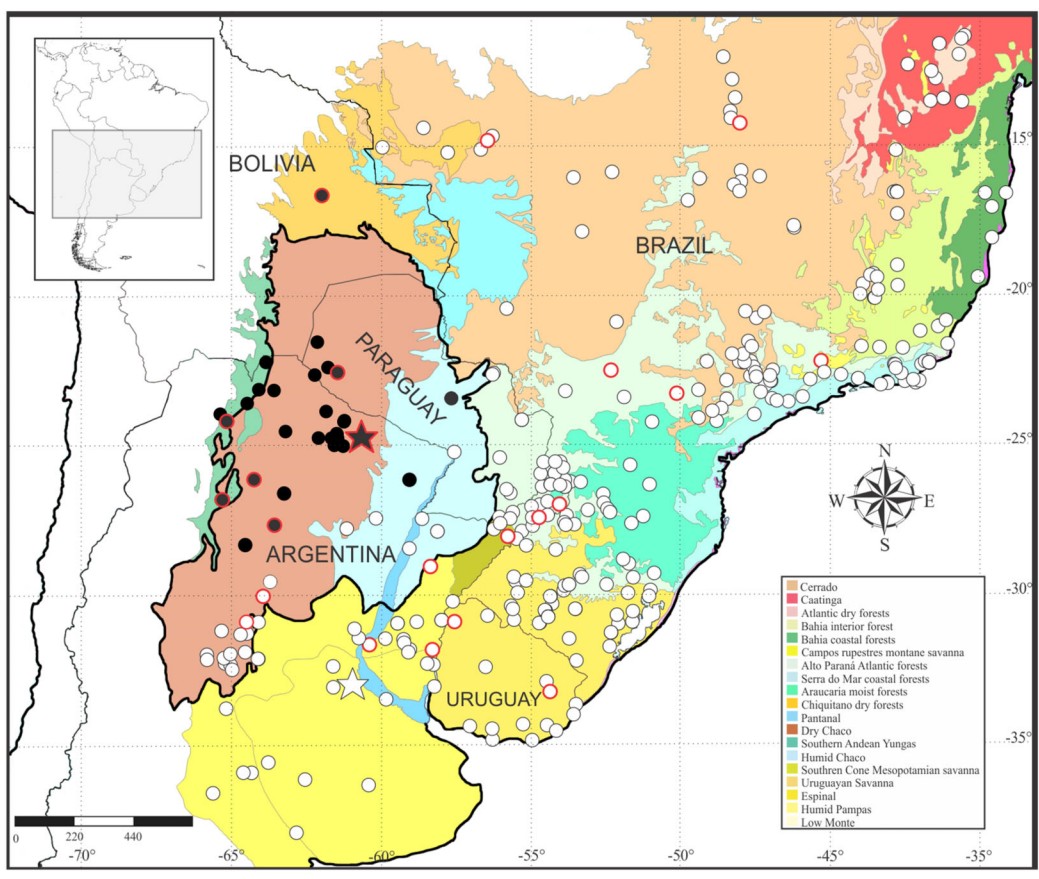

**Figure 8 Geographic distribution of *Leptodactylus apepyta* sp. nov. and *L. mystacinus*.** Localities based on examined specimens (Appendix II). Black circles: *L. apepyta* sp. nov., black star indicates its type locality (Las Lomitas, Formosa province, Argentina); white circles: *L. mystacinus*, white star indicates its type locality (Rosario, Santa Fe province, Argentina). Red borders on circles show sequences used in molecular analysis. Area enclosed in black denotes the limits of the South American Gran Chaco. South American ecoregions modified from *Olson et al. (2001)*.

Bolivia), was recovered as sister of the all remaining terminals of *L. apepyta* sp. nov. in the phylogenetic analyses (Fig. 1B), and due to its low genetic distance (0.8–1%) we consider it conspecific with *L. apepyta* sp. nov. Therefore, the distribution range of *L. apepyta* sp. nov. would extend from middle Bolivia eastwards to the Paraguay River in Paraguay (Presidente Hayes department as eastern limits), and more southwards reaches the northern half of Chaco and center of Santiago del Estero and southern Tucumán provinces in Argentina.

## DISCUSSION

### Monophyly of *Leptodactylus* and the *L. fuscus* group, and comments on some internal relationships

Although our molecular sampling was designed only to determine the phylogenetic position of *L. apepyta* sp. nov. and we did not include nuclear genes (i.e., rhodopsin), main discrepancies with the previous hypotheses produced by the recent phylogeny of *Leptodactylus* by *De Sá et al. (2014)* deserve some discussion. In the total evidence analysis

of *De Sá et al. (2014)* both the genus *Leptodactylus* and the *L. fuscus* group appear as monophyletic, but paraphyletic in the molecular-only analysis. The analyses of *De Sá et al. (2014)* recovered *L. fragilis* as sister taxon of the clade composed by *L. melanonotus* + Hydrolaterae + all remaining *Leptodactylus*. Our dataset recovered *Leptodactylus* and the *L. fuscus* group as monophyletic but low supported, with *L. fragilis* within a clade of this group that included *L. longirostris* + *L. poecilochilus*. The non-monophyly of *Leptodactylus* obtained by *De Sá et al. (2014)* could be attributed to the sequence of *L. fragilis*, apparently chimeric with a species of Hylini. Similar discrepancies in topologies attributed to accidental chimeras or contaminated sequences were detected, e.g., in *Eupsophus* (*Blotto et al., 2013*), Cycloramphidae (*Fouquet et al., 2013*), Ceratophryidae (*Faivovich et al., 2014*).

*Leptodactylus mystacinus* + *L. apepyta* sp. nov. were recovered by us in a clade formed also by *L. cupreus* (not included in *De Sá et al., 2014*), *L. troglodytes*, and *L. bufonius*, in a similar way to the total evidence results by *De Sá et al. (2014)* where *L. ventrimaculatus* + *L. labrosus* are the sister taxa of all other species of the *L. fuscus* group; with *L. bufonius*, *L. troglodytes*, and *L. mystacinus* forming a clade sister to the remaining species of the group. However, the molecular-only analysis of *De Sá et al. (2014)* resulted in a very different topology with *L. bufonius*, *L. troglodytes*, and *L. mystacinus* forming an early divergent grade within a clade with all species of *L. mystaceus* complex and some specimens of *L. fuscus*. In the description of *L. cupreus*, it was assigned to the *L. mystaceus* complex based on morphological and bioacustic characters (*Caramaschi, Feio & São-Pedro, 2008*). However, *Cassini et al. (2013)* argued that this relationship was only tentative and presented evidence suggesting that this species would be more similar to *L. mystacinus*, which is congruent with our results.

The terminals of *L. fuscus* 1–2 and 4–7 (we excluded sequences of *L. fuscus* 3, 8, and 9 for low quality or missing data) formed a clade with relatively low genetic distances; while in the analysis by *De Sá et al. (2014)* the terminals 6–9 were separately nested with *L. mystaceus*. *Camargo, De Sá & Heyer (2006)* affirmed that while their molecular data supports a multiple-species hypothesis for *L. fuscus*, they did not find relevant differences in calls and morphology to strongly clarify the taxonomic status of different lineages. Regarding *L. mystaceus*, we only used sequences of two lineages (1 and 3 or *L.* cf. *mystaceus*), which were sister taxa, contrary to *De Sá et al. (2014)* who found them as paraphyletic lineages. We do not rule out those discrepancies with our results, it could be due to the inclusion of contaminated and low-quality sequences in the analysis of *De Sá et al. (2014)*.

### Taxonomic conclusions

Phylogenetic topology, genetic *p*-distances, external morphology, coloration, osteology, and behavior support the view that the nominal species *L. mystacinus* is actually a complex composed by two clearly distinct taxa: *L. mystacinus*, widely distributed and with strong geographical and genetic structure, and *L. apepyta* sp. nov., endemic to the South American Gran Chaco. The 3% genetic divergence found in the 16S rDNA between both species is considered as "moderate" (*Vences et al., 2005*), but enough to identify candidate species among Neotropical anurans. The occurrence of cryptic species complexes is a frequent phenomenon in anurans (*Vences & Wake, 2007*; *Castroviejo-Fisher et al., 2017*). However, this phenomenon has been poorly studied in the specious genus *Leptodactylus*, with only

some conflicts being reported in *L. mystaceus* (*Heyer, García-Lopez & Cardoso, 1996*; *De Sá et al., 2014*), *L. fuscus* (*Wynn & Heyer, 2001*; *Camargo, De Sá & Heyer, 2006*), *L. gracilis* (*Silva et al., 2004*), and *L. pentadactylus* (*Gazoni et al., 2018*), among others. The moderate levels of genetic divergence between *L. mystacinus* and *L. apepyta* sp. nov., and its disjunct pattern ranges, could be explain that both species have a great morphological similarity and identical advertisement calls. The morphological characters we studied (i.e., general shape, head shape, tympanic annuli coloration, general coloration, and striped patterns) allowed us to unequivocally distinguish *L. apepyta* sp. nov. from *L. mystacinus*. Although it was not possible to include sequences from the type locality of *L. mystacinus* (Rosario, Santa Fe province, Argentina), external morphology of the specimens analyzed from nearby locations (Las Rosas and Los Nogales, Santa Fe province), and high quality photographs of the holotype of this species, allow easily to diagnose them as *L. mystacinus*, and distinguish them from *L. apepyta* sp. nov.

In addition to external morphology, we found few osteological differences between the two species in the shape of nasals and prevomers. *Ponssa (2008)* performed osteological studies on some populations of *L. mystacinus* from Argentina including specimens of *L. apepyta* sp. nov., but this author did not report relevant differences among them. Furthermore, a conservative osteological pattern can be depicted from the available osteological descriptions of species in the *L. fuscus* group, with low variation among them (*Heyer, 1998*; *Ponssa, 2006*, *2008*; *Ponssa & Barrionuevo, 2012*).

The advertisement calls of *L. mystacinus* analyzed in this study showed a structure similar to previous descriptions (*Barrio, 1965*; *Heyer, Heyer & De Sá, 2003*; *Oliveira Filho & Giaretta, 2008*). We only found differences with the call duration reported by *Barrio (1965)*, of 100 ms; however, *Heyer, Heyer & De Sá (2003)* suggested that the record of these high values could be due to the methodology employed. The advertisement calls of *L. apepyta* sp. nov. are structurally quite similar to the advertisement calls of *L. mystacinus*, as well as the quantitative parameters, being indistinguishable. This phenomenon has already been observed in complexes of cryptic species with allopatric distributions (*Heyer & Reid, 2003*; *Vences et al., 2008*; *Caminer et al., 2017*), while species with sympatric distributions tend to differentiate from each other, according to the character displacement hypothesis (*Brown & Wilson, 1956*), e.g. in *Heyer, García-Lopez & Cardoso (1996)*; *Funk, Caminer & Ron (2011)*.

The natural history of *L. apepyta* sp. nov. is poorly known and virtually nothing is known about its reproduction (e.g., amplexus, site and mode of oviposition, number of eggs, and tadpoles). Noticeably, males show a unique behavior among Leptodactylids; they were found vocalizing on the top of fallen logs and low branches of trees, up to 1.5 m high (Fig. 5B). This behavior was never reported neither for its close relative *L. mystacinus* (*Barrio, 1965*; *Langone, 1995*; *Abrunhosa, Wogel & Pombal, 2001*; *Heyer, Heyer & De Sá, 2003*; *Oliveira Filho & Giaretta, 2008*) nor for any other *Leptodactylus* species or genus in Leptodactylidae (*Lynch, 1971*; *Heyer, 1978*; *De Sá et al., 2014*). Reproductive males of *L. mystacinus* vocalize at shelter under dead logs or rocks, and males of most species in the *L. fuscus* group usually call from the ground, near or inside underground chambers that they construct for reproduction (*Heyer, 1978*; *De Sá et al., 2014*).

## Conservation status

According to the criteria of the International Union for Conservation of Nature, *L. apepyta* sp. nov. should be considered in the category "Least concern." However, the known distribution of *L. apepyta* sp. nov. corresponds to a new endemism for the Gran Chaco ecoregion and eventually also to the Chiquitanía ecoregion, which is a transitional zone between the Chaco and the Amazon with prevailing characteristics of Chaco environments (*Ibisch, Columba & Reichle, 2002*). The South American Gran Chaco is considered as of Conservation Priority 1 (Highest Priority at Regional Scale; *Dinerstein et al., 1995*). Although it has been considered a region with low diversity at the species level for some taxonomic groups (*Short, 1975*; *López-González, 2004*), it is inhabited by a large number of species of anurans, including two genera and more than 20 endemic species (*The Nature Conservancy, 2005*; *Nori et al., 2016*). In addition, this region represents the largest continuous dry forest remnant in South America, evidencing its relevance for future conservation efforts (*Maldonado & Hohne, 2006*; *Gasparri & Grau, 2009*).

## ACKNOWLEDGEMENTS

We thank to J. Faivovich and S. Nenda (MACN), S. Castroviejo-Fisher and G. Funk Pontes (PUCRS), M. Borges Martins and D. Janisch Alvares (UFRGS), N. Pupin (CFBH), T. Grant (MZUSP), S. Cechin (UFSM), M.T. Rodrigues (USP), J. Pombal Jr. and M. Woitovicz Cardoso (MNRJ), and D. Arrieta (MNHN) for providing access to specimens and assistance during the visits and work in the respective herpetological collections; to the collectors of the studied material; to E.R. Krauczuk, M. Sánchez, and A.S. Du Port Bru for assistance with photographs; to S. Cairo, R. Herrera, F. Marangoni, M.L. Ponssa, Fonoteca Neotropical Jacques Vielliard (FNJV, UNICAMP), Fonoteca Zoológica of the Museo Nacional de Ciencias Naturales (FonoZoo, Madrid) and EcoRegistros (Argentina) for providing recordings of advertisement calls.

### Funding

This work was supported by the Agencia Nacional de Promoción Científica y Tecnológica (PICTs 2381/2015, 2437/2017), the São Paulo Research Foundation (FAPESP Procs. # 2013/50741-7, # 2014/50342-8), the Conselho Nacional de Desenvolvimento Científico e Tecnológico, the Consejo Nacional de Ciencia y Tecnología (PRONII, CONACYT, Paraguay), and the Agencia Nacional de Investigación e Innovación (ANII/SNI, Uruguay). The funders had no role in study design, data collection and analysis, decision to publish, or preparation of the manuscript.

### Grant Disclosures

The following grant information was disclosed by the authors:
Agencia Nacional de Promoción Científica y Tecnológica: PICTs 2381/2015, 2437/2017.
São Paulo Research Foundation: FAPESP Procs. # 2013/50741-7, # 2014/50342-8.

Conselho Nacional de Desenvolvimento Científico e Tecnológico, the Consejo Nacional de Ciencia y Tecnología (PRONII, CONACYT, Paraguay).
Agencia Nacional de Investigación e Innovación (ANII/SNI, Uruguay).

## Competing Interests

The authors declare that they have no competing interests.

## Author Contributions

- Rosio G. Schneider conceived and designed the experiments, performed the experiments, analyzed the data, contributed reagents/materials/analysis tools, prepared figures and/or tables, authored or reviewed drafts of the paper, approved the final draft.
- Dario E. Cardozo performed the experiments, analyzed the data, contributed reagents/materials/analysis tools, authored or reviewed drafts of the paper, approved the final draft.
- Francisco Brusquetti analyzed the data, contributed reagents/materials/analysis tools, authored or reviewed drafts of the paper, approved the final draft.
- Francisco Kolenc conceived and designed the experiments, contributed reagents/materials/analysis tools, authored or reviewed drafts of the paper, approved the final draft.
- Claudio Borteiro conceived and designed the experiments, contributed reagents/materials/analysis tools, authored or reviewed drafts of the paper, approved the final draft.
- Célio Haddad conceived and designed the experiments, contributed reagents/materials/analysis tools, authored or reviewed drafts of the paper, approved the final draft.
- Nestor G. Basso conceived and designed the experiments, contributed reagents/materials/analysis tools, authored or reviewed drafts of the paper, approved the final draft.
- Diego Baldo conceived and designed the experiments, analyzed the data, contributed reagents/materials/analysis tools, prepared figures and/or tables, authored or reviewed drafts of the paper, approved the final draft.

## Field Study Permissions

The following information was supplied relating to field study approvals (i.e., approving body and any reference numbers):

Specimen collections were made in each country with the following authorizations numbers: Argentina (MEyRNR 007/2009, 048/2013, 072/2014, 061/2015, 073/2016, 035/2017, DPB 171/2015), Brazil (SISBIO 57098-1), Paraguay (MADES 196/2018, 232/2017, 186/2016); and Uruguay (MGAyP 199/13 and 137/16).

## DNA Deposition

The following information was supplied regarding the deposition of DNA sequences:

The new sequences are available at GenBank: MN153827–MN153849 (12S partial sequences) and MN153966–MN153988 (16S partial sequences).

## Data Availability

The raw data are available in the Supplemental Files: morphological raw data, bioacoustical raw data, and sound files.

## New Species Registration

The following information was supplied regarding the registration of a newly described species:

Publication LSID: urn:lsid:zoobank.org:pub:7AE8062D-2F27-4F99-AC5E-8D2A94874EF3.

Leptodactylus apepyta sp. nov. LSID: urn:lsid:zoobank.org:act:BCA45181-9C6B-4275-A9E9-BA58D96DE5CE.

## Supplemental Information

Supplemental information for this article can be found online at http://dx.doi.org/10.7717/peerj.7869#supplemental-information.

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
