# Peer review of "A new frog of the Leptodactylus fuscus species group (Anura: Leptodactylidae), endemic from the South American Gran Chaco"

_PeerJ, doi:10.7717/peerj.7869_

## Round 0.1 · original submission · Minor Revisions

Both reviewers and myself found your study to be an impressive and thorough piece of work documenting a new Leptodactylus lineage.

Reviewer 1 makes useful suggestions for enhancing the content in your abstract. I would also agree with Reviewer 1's suggested changes in response to the use of 'lineage' versus 'clade'. A more thorough discussion of the reddish dorsum in this new lineage as comapred to closely-related species would also be useful to include. Reviewer 2's comments about denoting more detail on your maps is important to address as well.

I have a few additional comments of my own.

First, please begin your methods with a description of the specimens used for the molecular methods and the extent to which these were recently collected by yourselves or sourced from other collections.

Lines 162-164: did you dissect the adults to determine ovarian follicles? If so, why not confirm the sex of all individuals by examining the gonads rather than relying on perhaps unreliable secondary sexual characteristics?

Would the authors be willing to put their morphological and call variables into a principle components analysis to further distinguish apepyta sp. nov. from mystacinus? It would be interesting to see which measurements load most strongly on the axes that explain the most variation in these morphological data.

Congratulations on this fine piece of work. We look forward to seeing a revised version.

Reviewer 1 ·

Basic reporting

Abstract:
The abstract could be revised to reflect some additional points that is already present in the body of the text (results and discussion).

1. For e.g. line 26. Advertisement calls can be mentioned in the list of criteria examined. Behavior is mentioned in the introduction (Line 83). Consider replacing multiple character system with one of the established terminology.
2. Also, the results pertaining to the phylogenetic position of L. mystacinus, recovered as sister in their analysis – one of the crucial findings – could appear in the abstract.
3. Finally, if authors prefer, they could insert a statement of comparison between the new lineage and its extant sister.
4. Line 31-32: consider paraphrasing “males call on tree branches.” Also check whether it is a unique habit (according to results it also calls on logs)
5. The main cephalic index character that separates this new lineage from its sister could come in the abstract.


Introduction:

Paragraph 1:
Line 36: consider removing of before Country names.
Line 40 – 41, contains some repetition on the geographical coverage that has already been mentioned in the first sentence (Line 35 – 39).
Line 68: Since you start the introduction with the definition of Gran Chaco region, mention the species range within this region might suffice.



Results:

Line 213: Since you are dealing with individual lineages here, consider using the term “lineages” instead of “clades”.
Line 219 – 221: Exact p-distance with the focal species of comparison (L. mystacinus) might help.
Line 247: “lineages or species” instead of “clades” might be appropriate here.
Line 409: In the figure 9, the individual in 1st row* 3rd column, doesn’t possess reddish brown dorsum. Similarly, in figure 10, the individual in 3rd row*2nd column shows reddish brown dorsum. This suggests coloration might be an overlapping character for the two lineages.
Line 411 – 414: It could be helpful, if the skeletal parts mentioned are named in the Fig 6 and 7. For example, the lateral edges of the middle ramus of prevomers can be labelled in the figures.
Line 445: comma usage
Line 447: “teeth present” instead of “teeth presents”
Line 449. To maintain consistency consider including the ratio in closed brackets.

Discussion
Line 603: It is unclear what they mean by terminals (1-2 and 4-7).
Line 620: they consider it as a cryptic lineage. Considering the moderate levels of divergence and potential geographical isolation, this pattern observed could be expected. Authors could consider expanding on this.
Line 627: The general shape alluded here as an unequivocal character needs to be mentioned in the result.



Reference:
695: 2001 (bold)
Line 1066: check Sabaj 2019 -> Sabaj 2016
Line 1048: specie -> species
Line 719: Baker MR, Vaucher C. 1984. (check Line 265: it says 1986)
Line 267: de la Riva (2000) -> de la Riva et al. (2000)
Line 842: Gallardo JM, de Olmedo EV. 1991. 
 check Line 269:Gallardo & Varela de Olmedo (1992) 

Line 968 and Line 970: one of the citations the year could be different. See line 273-274
Line 976 – 980: a & b missing against year 2000.
Line 1046: Pinto-Viveros et al. 2017 (check line 276: Pinto-Viveros et al. 2015)
Line 973: Lavilla and Manzano 1993 (check line 280: Lavilla and Manzano 1995)
Line 881: Heyer et al. 2005 (check line 389: Heyer et al. 2006)
Line 399: Heyer et al. 2006 (as above)
Line 402: Heyer 1996 missing
Line 729: Blotto et al. 2013 (Check line 588 Blotto et al. 2012)
Line 1054: year 2012 (check line 639 – 640 year 2010)
Line 738: year 1956 (check like 650 – year 1968)
Line 651: de Carvalho & Giaretta (2013) missing
Line 813: missing in the text
Line 1090: Swofford 2000 (check line 140, Swofford 2002)

Tables and Figures
Table 3: caption: consider using “individuals” instead of “specimens”
Fig 1: It is unclear what the colors on the branches refer to. Also, please provide nodal support values.
Figure 11: Most of the ecoregions plotted on the map might not be relevant for the paper. Highlighting the Gran Chaco (dry vs humid) might help improve the readability of the map. Also, a few reference localities could help in understanding the text.

Experimental design

Materials and methods:
Line 91 -92: It is unclear why two different sets of primers were used for both 12S and 16S.
Line 107 -108 – was it a single sequence (KM091585) or, multiple sequences? Consider revising the line accordingly.
Line 112: These appear like part of the ingroup. Please check.
Line 121 – 122: Here its mentioned as Bayesian approach while in the introduction (line 84) it appears as Bayesian criteria. Maintain consistency.
Line 128: use an alternative word: “studied”
Line 149-150: it is unclear: “currently deposited in the MACN collection”
Line 178: Please check the model number of directional microphone: Sennheiser LR 6?

Validity of the findings

Line 204 – 205. According to the Fig1a, there is no nodal support for this statement. The information provided in the main figure is insufficient to evaluate the inferences provided by the authors. Since, the focal new species along with its sister form a well-supported clade, the above issue is unlikely to affect the overall outcomes of the study. Nevertheless, the authors should update Fig 1. I could see that their Bayesian tree in the appendix shows appropriate support values.

Additional comments

First, I thank for the opportunity to review this manuscript. “A new frog from the Leptodactylus fuscus species group (Anura: Leptodactylidae), endemic from the South American Gran Chaco” by Schneider et al. The research is based on an extensive dataset (>780 individuals for morphological comparison) and multiple-criteria including phylogenetic analysis, external morphology, osteology and, advertisement calls, to discover, describe, and diagnose a new lineage of frog in the Gran Chaco region of Argentina. I congratulate the authors for this work.

·

Basic reporting

This paper is very clearly written, detailed, and adheres to the journal’s format.

Experimental design

This is a taxonomic paper. They hypothesize the presence of a new species (Leptodactylus apepyta sp. nov.) within the L. fuscus group and they support their hypothesis with several different lines of data. They use well established molecular and morphological methods to confirm the existence of this new species and bolster their findings with additional data like call and behavioral data. Overall the paper is extremely thorough and uses high sampling across the species group distribution in order to establish the full variation and distribution of L. apepyta sp. nov.

Validity of the findings

The authors were very thorough in evaluating each line of evidence in order to establish this new species. In addition they follow ICZN rules to insure that the species name will be valid according to the code.

Additional comments

This is a solid species description. I think it would be helpful to denote on the map which samples were sequenced for this description. Are these samples well distributed across both species ranges? You provide locality data for L. apepyta sp. nov. in Appendix 1 and you should provide similar detail for all of the sister taxon, L. mystaceus.

Minor Notes
Line 126, PartitionFinder is one word.
Line 481: Variation. Discuss variation of dorsolateral folds (1 vs 2) within L. apepytat. Is it random? Do most of 1 fold or 2? This features seems important for distinguishing species so it may be good to discuss this.
Line 435, remove period following (Fig. 8C,D)
Line 549: Natural History notes. Please describe the habitat in a bit more detail. Are the males calling over water? If so is this running water like a stream? Or are these ponds? Are the permanent bodies of water or ephemeral?

---

## Round 0.2 · accepted · Accept

Your group has put a tremendous amount of effort into the original manuscript and this revised version. I appreciate your thoughtful consideration of the reviewers' comments - the manuscript represents a robust contribution of a new species description.